# MS-SSM: A Multi-Scale State Space Model for Efficient Sequence Modeling

**Mahdi Karami** [1], **Ali Behrouz** [1], **Peilin Zhong** [1], **Razvan Pascanu** [2], **Vahab Mirrokni** [1]
[1]Google Research, [2]Google DeepMind
{mahdika,alibehrouz,peilinz,razp,mirrokni}@google.com

## Abstract

State-space models (SSMs) have recently attention as an efficient alternative to computationally expensive attention-based models for sequence modeling. They rely on linear recurrences to integrate information over time, enabling fast inference, parallelizable training, and control over recurrence stability. However, traditional SSMs often suffer from limited effective memory, requiring larger state sizes for improved recall. Moreover, existing SSMs struggle to capture multi-scale dependencies, which are essential for modeling complex structures in time series, images, and natural language. This paper introduces a multi-scale SSM framework that addresses these limitations by representing sequence dynamics across multiple resolution and processing each resolution with specialized state-space dynamics. By capturing both fine-grained, high-frequency patterns and coarse, global trends, MS-SSM enhances memory efficiency and long-range modeling. We further introduce an input-dependent scale-mixer, enabling dynamic information fusion across resolutions. The proposed approach significantly improves sequence modeling, particularly in long-range and hierarchical tasks, while maintaining computational efficiency. Extensive experiments on benchmarks, including Long Range Arena, hierarchical reasoning, time series classification, and image recognition, demonstrate that MS-SSM consistently outperforms prior SSM-based models, highlighting the benefits of multi-resolution processing in state-space architectures.

## 1 Introduction

Over the past few decades, numerous deep neural network architectures have been developed for sequence modeling. Early approaches like recurrent neural networks (RNNs) (Elman, 1990) and their variants, such as Long Short-Term Memory (LSTM) networks (Hochreiter et al., 1997) and Gated Recurrent Units (GRUs) (Cho et al., 2014), were proposed to handle sequential dependencies by maintaining hidden states over time. However, these models struggled with long-range dependencies and computational inefficiencies. With the advent of attention mechanisms (Bahdanau et al., 2015; Vaswani et al., 2017), the Transformer architecture emerged as the *de facto* standard for many sequence modeling tasks. The Transformer's self-attention mechanism enabled the modeling of complex relationships across sequences without relying on recurrence, allowing for parallel computation and better handling of long-range dependencies which enabled breakthrough advances across a wide range of applications. However, inference in transformer can be expensive due to the quadratic complexity of the attention mechanism, hindering its ability to handle even longer context tasks efficiently or run in low resource settings. These limitations has motivated the exploration of alternative scalable sequence modeling approaches with comparable expressiveness.

Recently, state-space models have generated renewed interest as efficient attention-free sequence models. Deep state-space models (SSMs), a class of RNNs that use linear recurrences, provide scalable training and inference capabilities, proving particularly effective for long-range dependency modeling (Gu et al., 2020a). These methods typically rely on a block structure similar to transformers, where the linear recurrences do sequence mixing,

while MLPs are used for feature mixing (Orvieto et al., 2023). To gain expressivity, similar to transformer, many such blocks are typically stacked on top of each other (Orvieto et al., 2024). The linearity allows to reformulate the recurrence as a convolution (Gu et al., 2020a; 2022a; 2021b; Mehta et al., 2022) or the use of associative scan (Smith et al., 2023; De et al., 2024), making SSM on par to transformer in terms of training cost. Recent architectures also typically use gating mechanisms, similar to LSTMs and GRUs, which can also be viewed as relying on input-dependent model parameters, increasing their expressivity (Gu & Dao, 2023; Orvieto et al., 2023; Dao & Gu, 2024; De et al., 2024; Beck et al., 2024), along with long convolution models (Karami & Ghodsi, 2024). They demonstrate considerable potential in various applications, including natural language processing (Gu & Dao, 2023; Karami & Ghodsi, 2024), computer vision (Liu et al., 2024; Karami & Ghodsi, 2024; Behrouz et al., 2024a), DNA modeling (Nguyen et al., 2024; Gu & Dao, 2023), and graph data (Behrouz & Hashemi, 2024).

However, traditional SSMs lack the inherent ability to capture multi-scale patterns prevalent in many real-world signals, such as image, audio, and time series data. Moreover, the *effective memory* of these linear RNNs, which is inversely proportional to the distance of the eigenvalues from the unit circle (Agarwal et al., 2023), is limited, requiring larger state sizes for improved recall. To address these limitations, we propose incorporating Multi-Resolution Analysis (MRA) into SSMs. By decomposing the input sequence into multiple scales, our approach allows the SSM to capture both fine-grained details and broader trends simultaneously. This multi-scale representation enables SSM to effectively capturing historical patterns at multiple levels of granularity.

Multi-resolution analysis plays a crucial role in understanding and modeling complex patterns across diverse datasets, including audio (Van Den Oord et al., 2016), images (Long et al., 2015), time series (Deznabi & Fiterau, 2023), graph generation (Karami, 2024), and text (Tamkin et al., 2020; Tai et al., 2015; Bowman et al., 2016). The importance of this approach stems from the multi-scale properties inherent in these data types, where patterns and structures manifest at various levels and timescales. For instance, natural language data exhibit multi-scale patterns ranging from subword to word, phrase, sentence, paragraph, and document levels. Similarly, the multi-scale structure of images and videos can reveal details from pixel-level to higher-level scene interpretation. Recently evidence from neuroscience further underscores the significance of multi-resolution analysis, particularly in language processing. Specifically, Caucheteux et al. (2023) provide evidence supporting hierarchical predictive coding in language, showing that the human brain predicts speech in a hierarchical manner, with different brain regions responsible for different levels of prediction. This aligns with earlier observation that the brain continuously predicts a hierarchy of representations across multiple timescales in the cortical hierarchy (Wacongne et al., 2011). Consequently, modern language models augmented with hierarchical predictions across multiple timescales can improve their alignment with human brain responses. Furthermore, even in data without explicit multi-scale characteristics, this modeling approach can efficiently capture long-range dependencies (Shi et al., 2023).

Several approaches have been proposed to incorporate multi-resolution analysis into sequence modeling. For instance, Nawrot et al. (2021) introduce a hierarchical Transformer architecture that processes information across multiple levels of abstraction in language modeling tasks. This approach explores various strategies for downsampling and upsampling activations in Transformers, achieving efficient computation and improved performance on various benchmarks. The Clockwork RNN (Koutnik et al., 2014) enhances traditional RNNs by partitioning the hidden layer into modules that operate at different temporal frequencies. This structure allows for a more efficient processing of sequences with varying temporal dynamics, thereby improving performance on complex tasks. In the context of Fourier-based multiresolution models, techniques such as FNet (Lee-Thorp et al., 2021), Prism (Tamkin et al., 2020), and Orchid (Karami & Ghodsi, 2024) operate in both the spatial and frequency domains. However, these methods are inherently non-causal, as the Fourier transform is applied across the entire sequence, also Fourier transform is poor in time localization of the representation in the frequency domain. Shi et al. (2023) proposed a multi-resolution convolution as an efficient pattern memorization, utilizing learned convolution kernels with dilations shared across multiple timescales. However, similar to

other short convolution-based architectures, this model's effective receptive field is limited. Additionally, Fan et al. (2024) has utilized the intrinsic granularity present in data to design more stable and accurate forecasting methods using diffusion.

In this work, we introduce *MS-SSM*, which integrates an efficient multi-resolution analysis into the state space architecture, decomposing the dynamical system into multiple time scales. This enables the overall SSM to operate at different resolutions. We show the effectiveness of our methods on Long Range Arena (Tay et al., 2020b) as well as other sequential tasks. In section 2 we describe in detail the proposed method, providing our empirical evaluation in section 3.

## 2   Method

The proposed sequence model is composed of two core components: 1) a multi-scale decomposition and 2) an array of state space models (SSMs). These components work together to capture patterns and temporal dynamics at different resolutions. Each will be explained in detail in the following sections.

### 2.1   State Space Models

**SSM.**   State Space Models (SSMs) are linear time-invariant systems that map input sequence $x(t) \in \mathbb{R}^L$ to response sequence $y(t) \in \mathbb{R}^L$ (Aoki, 2013) using a latent state $h(t) \in \mathbb{R}^{N \times L}$, parameter $\mathbf{A} \in \mathbb{R}^{N \times N}$ (a.k.a. *state transition matrix*), and projection parameters $\mathbf{B} \in \mathbb{R}^{N \times 1}, \mathbf{C} \in \mathbb{R}^{1 \times N}$. That is: $h'(t) = \mathbf{A}\,h(t) + \mathbf{B}\,x(t), \quad y(t) = \mathbf{C}\,h(t)$. Discrete space state models (Gu et al., 2020a; Zhang et al., 2023) is obtained by discretizing at step size $\mathbf{\Delta}$ through a high accuracy Zero-Order-Hold (ZOH) method:

$$h_t = \bar{\mathbf{A}}\,h_{t-1} + \bar{\mathbf{B}}\,x_t \tag{1}$$
$$y_t = \mathbf{C}\,h_t,$$

where $\bar{\mathbf{B}} = (\mathbf{\Delta A})^{-1}\left(\exp\left(\mathbf{\Delta A} - I\right)\right).\mathbf{\Delta B}$ and $\bar{\mathbf{A}} = \exp\left(\mathbf{\Delta A}\right)$.

These models can be interpreted as both CNNs and RNNs and are equivalent to the convolution $\bar{\mathbf{K}} = \left(\mathbf{C}\bar{\mathbf{B}}, \mathbf{C}\bar{\mathbf{A}}\bar{\mathbf{B}}, \dots, \mathbf{C}\bar{\mathbf{A}}^{L-1}\bar{\mathbf{B}}\right)$, and so $y = x * \bar{\mathbf{K}}$ (Gu et al., 2020a). Leveraging the convolution theorem and Fast Fourier Transform (FFT) algorithm for this long convolution formulation, its training complexity scales quasi-linearly with sequence length and can be parallelized, while it enjoys linear complexity at inference time using its recurrence form.

Structured SSM (Gu et al., 2022a) relies on a diagonal parametrization of $\mathbf{A}$, enabling efficient computation of the discretization in (1) and its convolution formulation. Combined with the use of associative scan techniques Smith et al. (2023), this allows for efficient parallelization of computation even when using the recurrent form. Newer architectures such as Mamba (Gu & Dao, 2023) or Griffin (De et al., 2024), typically have moved away from the convolutional formulation.

**Input-Dependent SSM.**   Recently, Gu & Dao (2023) introduced the S6 block, a structured State Space Model (SSM) with a selective scan mechanism. This input-dependent gating mechanism enables S6 to selectively propagate or forget information along the sequence dimension by allowing specifying the parameters as:

$$\bar{\mathbf{B}}_t = s_B(x_t) = \text{Linear}_{\mathbf{B}}(x_t), \quad \mathbf{C}_t = s_C(x_t) = \text{Linear}_{\mathbf{C}}(x_t), \quad \mathbf{\Delta}_t = s_{\Delta}(x_t) = \text{Softplus}\left(\text{Linear}_{\mathbf{\Delta}}(x_t)\right),$$

where $\text{Linear}(\cdot)$ is a linear projection and $\text{Softplus}(\cdot) = \log(1 + \exp(\cdot))$. This approach adds context-awareness to SSMs and a similar form is used in other works, e.g. (De et al., 2024). Despite its more expressive power, in contrast to S4, this time- and input-variant model prevents the use of the convolutional formulation. But as mentioned above, computation can still be parallelized by using the *associative scan* (Martin & Cundy, 2018; Smith et al., 2023; Orvieto et al., 2023). Also, it allows for more hardware-aware implementations.

**Limitations:** While the linear formulations of SSM allows to greatly improve scalability of the system and to control its stability (Orvieto et al., 2023), it also limits the architecture. From

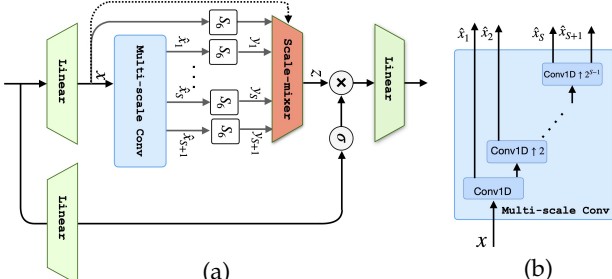

Figure 1: (a) Block diagram of the MS-SSM model. (b) Multi-scale convolution layer. The `multi-scale conv` layer, which decomposes the signal into multiple scales, is composed of nested convolution layers `Conv1d` defined in (3). The `scale-mixer` combines the scales through an input-dependent weighted summation defined in (4).

an expressivity point of view, a single linear recurrent layer is limited in what it can represent. Deep SSM architectures recapture expressivity by stacking multiple blocks Orvieto et al. (2024). Additionally, the system can only exhibit fading memory, where the time to live for information is inversely proportional to the distance of the eigenvalues from the unit circle (Agarwal et al., 2023), requiring an increase in the state size in order to improve the ability of the system to recall.

## 2.2 Multi-Scale Decomposition

Multi-resolution analysis (MRA) is a mathematical framework that enables the analysis of signals at multiple scales or resolutions. A powerful tool for performing MRA is the Discrete Wavelet Transform (DWT), which decomposes a signal into different levels of approximation and detail by recursively applying a pair of filters—a low-pass filter and a high-pass filter, denoted by $\varphi$ and $\psi$, respectively—followed by downsampling.[1]

A major limitation of the standard Discrete Wavelet Transform (DWT) is its lack of translation-invariance, meaning that even small shifts in the input signal can result in significant changes to the resulting wavelet coefficients. To address this issue, several DWT variants have been developed that use redundant signal representations. One such approach is the Dual-Tree Complex Wavelet Transform (DTCWT) (Selesnick et al., 2005), which provides approximate translation-invariance by using two parallel DWT trees with slightly different filters. In contrast, the Stationary Wavelet Transform (SWT) (Nason & Silverman, 1995) achieves true translation-invariance by skipping the downsampling step at each decomposition level. Given an input signal $a^0 \triangleq x$, the SWT decomposes it recursively into approximation and detail coefficients at each scale $s \in \{1, 2, ..., S\}$, as follows:

$$a^s[t] \triangleq (a^{s-1} * (\varphi \uparrow 2^{s-1}))[t] = \sum_{\ell=0}^{K-1} a^{s-1}[t - 2^{s-1}\ell]\varphi[\ell]$$

$$d^s[t] \triangleq (a^{s-1} * (\psi \uparrow 2^{s-1}))[t] = \sum_{\ell=0}^{K-1} a^{s-1}[t - 2^{s-1}\ell]\psi[\ell]. \tag{2}$$

In essence, the coefficients at level $s$ are obtained by convolving the upsampled filters, $(\varphi \uparrow 2^{s-1})$ and $(\psi \uparrow 2^{s-1})$, with the approximation coefficients from the previous level, $a^{s-1}$. The complete multi-scale decomposition of the signal after $S$ levels consists of the set of detail coefficients at all scales, $(d^1[t], ..., d^S[t])$, along with the final approximation coefficients, $a^S[t]$, which together can perfectly reconstruct the original signal. This transformation of the signal provides information about both the frequency content and the time localization of the signal and also captures both the smooth, global trends and the fine-grained details, enabling a wide range of applications in signal processing. One key advantage of the SWT is that it maintains the same sequence length at each decomposition level, producing a redundant representation of the signal. This redundancy is key to achieving translation-invariance, which leads to significant performance improvements in applications such as signal denoising (Kumar et al., 2021), image resolution enhancement (Demirel & Anbarjafari, 2010), and feature extraction (Zhang et al., 2010). However, the trade-off for this improved performance is the increased computational cost and memory usage compared to the standard DWT.

---

[1]Continuous form of multi-scale analysis such as continuous wavelet transforms are normally discretized with a finite dyadic set $\{2^s\}_{s=1}^S$.

The specific form of the filters $\varphi$ and $\psi$ depends on the choice of wavelet basis. Different wavelet families, such as Haar, Daubechies, and Symlets, have distinct filter coefficients, resulting in different properties for the wavelet transform (Daubechies, 1992). While choosing an orthogonal wavelet basis ensures perfect reconstruction of the signal, this property is not always desirable in deep sequence modeling. As observed in recent research (Shi et al., 2023), employing trainable filter weights instead of fixed wavelet bases offers greater flexibility and model expressiveness. This approach enables the model to learn optimal filter coefficients for specific tasks, potentially leading to enhanced performance in a range of applications. The filtering operation at level $s$, as defined in equation 2, can be efficiently implemented using a causal depthwise 1D convolution (Conv1d) with two output channels, a kernel length of $K$, and a dilation factor of $2^{s-1}$. As a result, the input-output relationship in (2) can be specified by [2]

$$[a^s; \, d^s] = \texttt{Conv1d}(1, \, 2, \, L, \, 2^{s-1})[a^{s-1}].$$ (3)

In this model, the multi-scale block utilizes convolution kernels with dedicated weights for each scale.

This recursive process leads to a nested multi-scale decomposition block that transforms a 1-dimensional sequence into a set of sequences across different scales, which can be collected into a multi-dimensional representation vector, *i.e.* $x_t \in \mathbb{R} \mapsto \hat{x}_t \in \mathbb{R}^{S+1}$. Each dimension in this representation corresponds to a different resolution, capturing signal features from fine-grained details to coarse global trends, enabling analysis of the signal across varying levels of granularity. The higher the scale value, $s$—which corresponds to deeper levels in the recursion tree of (3)—the more coarse-grained the information represented at that scale. This follows the *recursive principle* (Pauwels et al., 1995), whereby larger values of $s$ result in increasingly blurred (less sharp) representations of an image (Worrall & Welling, 2019).

At each time scale $s$, the dilated convolution filter captures patterns of length up to $2^s \times K$, meaning that $\hat{x}_t^s$ represents local patterns within a limited window preceding the time index $t$. In other words, akin to the localized spectro-temporal representation in the Discrete Wavelet Transform, the scale components of $\hat{x}_t$, with limited number of scales, capture only recent local structures. However, for non-local patterns that span larger intervals, such as those found in auditory signals (Romero et al., 2020), it is essential to model long-range temporal correlations within each scale representation. To address this, we apply independent SSMs—which maintain a global receptive field—to each scale representation, as well as to the original signal, in order to capture the temporal dynamics within the scales. The proposed models, named *MS-SSM*, specializes distinct SSMs for different time scales. This setup results in an array of $(S+2)$ SSMs operating in parallel, with each SSM having a latent state size of $N$. Consequently, the effective latent state size per input channel becomes $(S+2)N$. To obtain comparable state dimension in the proposed model, we set this effective state size to match the recurrent state size of other models, thereby maintaining consistent latent dimensions across different architectures. Additionally, this SSM array can be implemented in parallel, making their overall computational complexity comparable to architectures operating at a single resolution. The MS-SSM block is illustrated in Figure 1.

**Initialization.** The eigenvalues of the state transition matrix ($|\lambda_i(\bar{\mathbf{A}})|$) play a critical role in determining the stability and memory capacity of State Space Models. To ensure stability in discrete SSMs, these eigenvalues must lie within the unit circle, while for continuous-time SSMs, the eigenvalues of $\mathbf{A}$ must be in the left half-plane. Eigenvalues of $\bar{\mathbf{A}}$ that are closer to 1 enhance the model's ability to capture long-range dependencies (Gupta et al., 2022; Orvieto et al., 2023). In essence, the *effective memory* of an SSM, which quantifies how long past information influences the present state, is inversely proportional to the distance of the eigenvalues from the unit circle. Formally, when eigenvalues satisfy $|\lambda_i(\bar{\mathbf{A}})| < 1 - \delta$ the effective memory is on the order of $\frac{1}{\delta}$ (Agarwal et al., 2023).

To balance between capturing long-range dependencies and maintaining different effective memory at each resolution, we employ a *scale-dependent initialization scheme*. Previous works

---

[2]In PyTorch, this operation can be simply realized with the following code: `torch.nn.Conv1d(1, 2, kernel_size=L, dilation=2**(s-1))`.

observed that real-valued SSMs can perform on par with or even outperform complex-valued counterparts (Ma et al., 2022; Gu & Dao, 2023), hence, we adopt a diagonal-structured recurrence matrix with real values.

For lower resolutions (higher value of $s$ in hierarchy), which contain coarse-grained information, we initialize the diagonal elements of $\bar{\mathbf{A}}$ with values closer to 1 to enhance the model's ability to capture long-range dependencies within these scales. In contrast, for

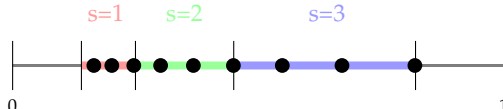

Figure 2: Initialization scheme for 3 different scales with $N = 3$ and $\Delta_0 = 0.2$.

higher resolutions containing fine-grained details, we initialize $\text{diag}(\bar{\mathbf{A}})$ with smaller values to prioritize shorter effective memory and focus on local dynamics at initialization. Specifically, the diagonal elements of the state transition matrix at scale $s \in \{0, \ldots, S+1\}$, $\text{diag}(\mathbf{A}^s)$, are initialized uniformly within the interval $\big(-N(S+1-s),\ -N(S-s)\big]$ (or equivalently $\text{diag}(\bar{\mathbf{A}}^s) \in \big(e^{(-\Delta_0 N(S+1-s))},\ e^{(-N\Delta_0(S-s))}\big]$ ), where $N$ is the state size per scale. By concatenating all latent states into a large state $[\mathbf{h}^0\ ;\ \ldots\ ;\ \mathbf{h}^{S+1}]$, the overall state transition matrix becomes $\mathbf{A} = \text{diag}([\text{diag}(\mathbf{A}^0)\ ;\ \ldots\ ;\ \text{diag}(\mathbf{A}^{S+1})])$. Then, this real-valued initialization aligns with that in the S4D-real (Gu et al., 2022a) which is grounded in the HiPPO theory (Gu et al., 2020a), where the $n$-th element of $\text{diag}(\mathbf{A})$ is initialized as $-(n+1)$. An example of this initialization scheme is illustrated in Figure 2.

**Scale Mixer.** After independently modeling the temporal dynamics at each specific scale, the array of $(S+2)$ SSMs produces outputs that are collected into the vector $\mathbf{y}_t \in \mathbb{R}^{S+2}$. To effectively merge these multi-scale representations, the model requires a mechanism that encodes cross-scale interactions, enabling information to flow between scales and ultimately combines them into a single-dimensional output. To achieve this, we combines the scales through a weighted summation applying an input-dependent projection matrix $\mathbf{E}_t \in \mathbb{R}^{1 \times (S+2)}$:

$$z_t = \texttt{scale-mixer}(\mathbf{y}_t; x_t) = \mathbf{E}_t\,\mathbf{y}_t, \quad \text{where } \mathbf{E}_t = s_E(x_t) = \text{Linear}_{\mathbf{E}}(x_t) \tag{4}$$

This approach allows the model to dynamically adjust the contribution of each scale based on its input.

**Input-dependent Parameterization.** In S6 (Gu & Dao, 2023), an input-dependent parameterization is employed for the SSM, allowing the model to selectively propagate or forget information along the sequence based on the input token of the SSM, functioning similarly to gating mechanism in RNNs. In this work, for the $s$-th SSM operating on scale $s$, we make the parameters functions of the original input $x_t$. Specifically, the parameters of the $s$-th SSM, are modeled as $\bar{\mathbf{B}}_t^s = s_B^s(x_t)$, $\bar{\mathbf{C}}_t^s = s_C^s(x_t)$, and $\Delta_t^s = s_\Delta^s(x_t)$. Through empirical studies, presented in Appendix C, we observe that gating based on the raw input, $x_t$, is more effective than gating based on the scale-specific representations ($\bar{\mathbf{B}}_t^s = s_B^s(\hat{x}_t^s)$, $\bar{\mathbf{C}}_t^s = s_C^s(\hat{x}_t^s)$, and $\Delta_t^s = s_\Delta^s(\hat{x}_t^s)$). Using the raw input for controlling the parameters results in a more effective mixing of each scale's representation with the raw input information.

**Complexity.** The multi-scale convolution operation introduces a linear time computation overhead of $\mathcal{O}(LKS)$ and require an additional $\mathcal{O}(KS)$ parameters per layer. However, this overhead is minimal compared to the overall model size, given the small convolution kernel size, $K$, and the limited number of scales, $S$.

# 3 Experiments

We evaluate our proposed architecture across image classification tasks, where images are converted into a sequence of patches (ImageNet-1k) or pixels (sCIFAR), as well as hierarhical reasoning and time series classifications. In all experiments, we report the results of two variants of our approach, i.e., MS-SSM (S4) and MS-SSM (S6), in which we use S4 (Gu et al., 2021a) and S6 (Gu & Dao, 2023) blocks as the recurrent module, respectively. Comparison of these two, as two instances of data-dependent and data-indpendent recurrent models, shows that MS-SSM'performance does not rely on the S6 block and supports the significance of our design.

Table 1: Results on sCIFAR (Shi et al., 2023) and ImageNet (Deng et al., 2009). Missing results mean that the performance of the model is not reported on ImageNet-1K in the original reference.

| Method | sCIFAR | ImageNet-1K |
|---|---|---|
| Transformers | | |
| Transformer (Vaswani et al., 2017) | 62.2 | 78.9 |
| Recurrent Neural Networks (RNNs) | | |
| HiPPO-RNN (Gu et al., 2020a) | 61.1 | - |
| LSTM (Hochreiter et al., 1997) | 63.0 | - |
| r-LSTM (Trinh et al., 2018) | 72.2 | - |
| UR-GRU (Gu et al., 2020b) | 74.4 | - |
| LipschitzRNN (Erichson et al., 2020) | 64.2 | - |
| State Space Models (SSMs) | | |
| S4 (Gu et al., 2022b) | 91.1 | 79.1 |
| S4D (Gu et al., 2022a) | 89.9 | 80.4 |
| S5 (Smith et al., 2023) | 89.7 | 77.9 |
| Liquid-S4 (Hasani et al., 2022) | 92.0 | - |
| Mamba (Gu & Dao, 2023) | 90.1 | 80.5 |
| Convolutions | | |
| CKConv (Romero et al., 2021) | 63.7 | - |
| MULTIRESNET (Shi et al., 2023) | 93.1 | - |
| Orchid (Karami & Ghodsi, 2024) | 93.0 | 80.2 |
| Convolution + SSMs | | |
| MS-SSM (S4) | 90.3 | 79.7 |
| MS-SSM (S6) | **93.3** | **81.3** |

Table 2: Performance of predicting outcomes of list operations in ListOps dataset of Tay et al. (2020b). *Mamba 2X Param* and *Mamba 2X State* denote Mamba model with double model size and double state size, respectively.

| Model | Accuracy (%) |
|---|---|
| Transformers | |
| Transformer (Vaswani et al., 2017) | 36.37 |
| Local Attention (Tay et al., 2020b) | 15.82 |
| Linear Trans. (Katharopoulos et al., 2020) | 16.13 |
| Linformer (Wang et al., 2020) | 16.13 |
| Sparse Transformer (Child et al., 2019) | 17.07 |
| Performer (Choromanski et al., 2020) | 18.01 |
| Sinkhorn Transformer (Tay et al., 2020a) | 33.67 |
| Longformer (Beltagy et al., 2020) | 35.63 |
| BigBird (Zaheer et al., 2020) | 36.05 |
| Luna-256 (Ma et al., 2021) | 37.25 |
| Reformer (Kitaev et al., 2020) | 37.27 |
| H-Transformer-1D (Zhu & Soricut, 2021) | 49.53 |
| Convolutions | |
| CDIL (Cheng et al., 2023) | 44.05 |
| SGConv (Li et al., 2022) | 61.45 |
| MULTIRESNET (Shi et al., 2023) | 62.75 |
| SSMs | |
| S4 (Gu et al., 2022b) | 59.60 |
| DSS (Gupta et al., 2022) | 57.60 |
| S4D (Gu et al., 2022a) | 60.52 |
| S5 (Smith et al., 2023) | 62.15 |
| Liquid-S4 (Hasani et al., 2022) | 62.75 |
| Griffin (De et al., 2024) | 32.34 |
| Mamba (Gu & Dao, 2023) | 38.02 |
| Mamba 2x Param | 49.63 |
| Mamba 2x State | 42.14 |
| Convolutions + SSMs | |
| MS-SSM (S4) | 62.83 |
| MS-SSM (S6) | **63.04** |

**Image Classification.** We evaluate the performance of MS-SSM in two image classification tasks: ImageNet-1K (Krizhevsky et al., 2012) and sCIFAR (Shi et al., 2023). We use ImageNet to compare the performance of MS-SSM with baselines in modeling the sequence of image patches. In sCIFAR task, however, each image is treated as a 1D sequence of pixel and so the models are not using any 2D inductive bias from the images. Therefore, the model must be able to capture long-range dependencies and patterns at different resolutions. Results are reported in Table 1. MS-SSM shows outstanding performance compared to all other sequence models in both tasks and more specifically in capturing long range and multi-resolution modeling of pixels in sCIFAR. The superior performance compared to Mamba (Gu & Dao, 2023) and similar SSM-based models (Smith et al., 2023; Gu et al., 2022b;a) comes from the multi-resolution convolutions that helps MS-SSM to capture the dependencies at different levels of granularity. Compared to multi-resolution methods, e.g., MULTIRESNET (Shi et al., 2023), the superior performance of MS-SSM highlights the significance of SSMs and our scale-mixer module.

**Time Series Classification.** Time series classification is one of the important tasks in sequence modeling that requires capturing dependencies at different resolutions. We use PTB-XL (Wagner et al., 2020), a commonly used dataset of electrocardiogram (ECG) in the time series literature. This dataset has 21,837 ECG recordings, each of which with 12 channels, from 18,885 patients. Each recording has at least one label from 71 total ECG labels obtained from SCP-ECG standard. In this experiment, the dataset is partitioned into six subsets of "all", "diagnostic", "diagnostic subclass", "diagnostic superclass", "form", and "rhythm". Following previous studies (Behrouz et al., 2024b; Shi et al., 2023), we use the 100Hz version of the dataset, in which each time series has 1000 timesteps. Table 3 reports the results on ECG classification tasks. MS-SSM outperforms all the baselines, even specialized models for time series (e.g., SpaceTime (Zhang et al., 2023)).

**Hierarchical Reasoning.** To evaluate the MS-SSM's ability in reasoning about hierarchical structures, we perform experiments on the long ListOps dataset from the Long Range Arena benchmark (Tay et al., 2020b). This dataset consists of sequences with hierarchical structures and operators such as MAX, MIN, MEDIAN, and SUM_MOD, which are enclosed by brackets to indicate nested operations. A short example of a sequence from this dataset is as follows:

Table 3: AUROC for ECG multi-label/multi-class classification on the PTB-XL dataset.

| Model (AUROC) | All | Diag | Sub-diag | Super-diag | Form | Rhythm |
|---|---|---|---|---|---|---|
| Transformer (Vaswani et al., 2017) | 0.857 | 0.876 | 0.882 | 0.887 | 0.771 | 0.831 |
| MULTIRESNET (Shi et al., 2023) | 0.938 | 0.939 | 0.934 | 0.934 | 0.897 | 0.975 |
| Spacetime (Zhang et al., 2023) | 0.936 | **0.941** | 0.933 | 0.929 | 0.883 | 0.967 |
| S4 (Gu et al., 2022b) | 0.938 | 0.939 | 0.929 | 0.931 | 0.895 | 0.977 |
| InceptionTime (Ismail Fawaz et al., 2020) | 0.925 | 0.931 | 0.930 | 0.921 | 0.899 | 0.953 |
| LSTM (Hochreiter et al., 1997) | 0.907 | 0.927 | 0.928 | 0.927 | 0.851 | 0.953 |
| Wavelet features (Strodthoff et al., 2020) | 0.849 | 0.855 | 0.859 | 0.874 | 0.757 | 0.890 |
| Mamba (Gu & Dao, 2023) | 0.915 | 0.929 | 0.905 | 0.912 | 0.876 | 0.952 |
| MS-SSM (S4) | **0.939** | 0.939 | 0.935 | 0.930 | 0.899 | **0.980** |
| MS-SSM (S6) | **0.939** | **0.941** | **0.936** | **0.935** | **0.901** | 0.979 |

Table 4: Performances Comparison on the Long Range Arena benchmark (Tay et al., 2020b). The baselines results are reported by Qin et al. (2024).

| Model | Text | Retrieval | Image | Pathfinder | Path-X | AVG. |
|---|---|---|---|---|---|---|
| Transformer (Vaswani et al., 2017) | 61.95 | 80.69 | 40.57 | 65.26 | - | 62.12 |
| cosFormer (Qin et al., 2022) | 67.70 | 83.15 | 51.23 | 71.96 | - | 68.51 |
| FLASH (Hua et al., 2022) | 64.10 | 86.10 | 47.40 | 70.25 | - | 66.96 |
| S4 (Gu et al., 2022b) | 86.82 | 90.90 | 88.65 | 94.20 | 96.35 | 91.38 |
| DSS_softmax (Gupta et al., 2022) | 84.80 | 87.80 | 85.70 | 84.60 | 87.80 | 86.13 |
| DSSEXP (Gupta et al., 2022) | 84.60 | 87.60 | 84.90 | 84.70 | 85.60 | 85.47 |
| DSSEXP-NO-SCALE (Gupta et al., 2022) | 82.40 | 86.00 | 81.20 | 81.30 | - | 66.46 |
| TNN (Qin et al., 2023) | 87.90 | 90.97 | 88.24 | 93.00 | 96.10 | 91.24 |
| S5 (Smith et al., 2023) | 89.31 | 91.4 | 88.00 | 95.33 | 98.56 | 92.52 |
| Mega (Ma et al., 2022) | 90.43 | 91.25 | 90.44 | 96.01 | 97.98 | 93.22 |
| SGConv (Li et al., 2022) | 89.2 | 91.11 | 87.97 | 95.46 | 97.83 | 92.31 |
| LRU (Orvieto et al., 2023) | 89.40 | 89.90 | 89.00 | 95.10 | 94.20 | 91.52 |
| Mamba (Gu & Dao, 2023) | 82.98 | 72.14 | 69.82 | 69.26 | 67.32 | 72.30 |
| Griffin (De et al., 2024) | 71.75 | 66.58 | 61.15 | 73.38 | 69.53 | 68.47 |
| MS-SSM (S4) | 87.22 | 91.06 | 89.15 | 94.90 | 97.12 | 91.89 |
| MS-SSM (S6) | 85.70 | 83.21 | 89.83 | 87.24 | 87.70 | 86.73 |

```
INPUT: [MAX 2 4 [MIN 1 6 ] 1 0 [MEDIAN 1 9 7]]      OUTPUT: 7
```

Table 2 reports the performance of MS-SSM and baselines on ListOps dataset. MS-SSM achieves the best results compared to all baselines. Notably, MS-SSM achieves ×2 accuracy compared to Mamba (Gu & Dao, 2023), which shows the significance of multi-resolution modeling of the sequence.

Additionally, the performance improvement is achieved without increasing computational complexity or parameter count. When compared to Mamba models with double parameter count and double state size, MS-SSM consistently exhibits superior performance, highlighting its effectiveness and efficient utilization of its multi-timescale memory in capturing long hierarchical structures.

**Long Range Arena.** We further evaluate the performance of MS-SSM on additional tasks from the Long Range Arena benchmark (Tay et al., 2020b). The results, summarized in Table 4, highlight the advantages of MS-SSM over similar data-dependent SSM-based architectures such as Mamba and Griffin. While these models exhibit poor performance on long-range tasks, MS-SSM achieves a significant 14.42% performance improvement over its closest counterpart, Mamba, which shares a similar SSM architecture.[3] This performance boost is attributed to the integration of multi-scale convolutions, which enhances MS-SSM's capacity to capture dependencies across various scales and over long sequences.

**Ablation Studies.** In this section, we evaluate the significance of our model design and the made choices by performing an ablation study on ListOps and PTB-XL datasets. To this end, we change the main components of the MS-SSM, one at a time, to evaluate its contribution in the performance of MR-SSM. We use the following variants: (1) is the main variant of MS-SSM, when using S6 block as the recurrent module, (2) re-

---

[3]The primary of this work goal is not to achieve SOTA results on LRA and other benchmarks. As it share similar SSM architecture with Mamba, a fair comparison is conducted against it.

places the S6 block with S4, (3) removes the recurrent module, (4) removes the multiresolution convolution and instead uses Conv1D, (5) is the original gating for scales, (6) for each scale, we use its own input for the gating, (7) is the gating where each scale is gated with the original input, (8) is the original scale mixing module used in MS-SSM, (9) uses simple linear layer for mixing different scales, and (10) uses non-linearity in the gating (data-dependency) of scale mixing. The results are reported in Table 5, indicating that all components contributes to the performance gain, where main contribution comes from the multiresolution convolution. Additional experimental results and ablation studies (on the types of initialization) are discussed in Appendix C.

Table 5: Ablation on the architecture of MS-SSM.

| | Method | PTB-XL | ListOps |
|---|---|---|---|
| | Base | | |
| 1 | MS-SSM (S6) | 0.939 | 63.04 |
| 2 | MS-SSM (S4) | 0.939 | 62.83 |
| 3 | Remove S6/S4 | 0.936 | 62.59 |
| 4 | Remove Multi. Conv. | 0.916 | 37.98 |
| | Gating (Input, Based on) | | |
| 5 | (self scale, original input) | 0.939 | 63.04 |
| 6 | (self scale, self scale) | 0.938 | 62.91 |
| 7 | (original input, self scale) | 0.939 | 62.95 |
| | Scale Mixing | | |
| 8 | Input-dependent | 0.939 | 63.04 |
| 9 | Input-independent | 0.932 | 61.28 |
| 10 | None-linear `SoftMax` gate | 0.921 | 61.42 |

## 3.1 Effective Receptive Field

We introduce the concept of the *mean mixing distance* as a metric to quantify the effective receptive field (ERF) in our model, drawing inspiration from the receptive field in convolutional networks. This definition is inspired by the average attention distance defined in self-attention models (Dosovitskiy et al., 2020).

The normalized attention scores between each pair of tokens defines the mapping between each output token and all tokens in the input sequence.[4] Using this, the average attention distance (Dosovitskiy et al., 2020) is defined as: $d(m, n) = \sum_{n=1}^{m} \mathbf{A}(x)_{m,n} \times (m - n)$ where each row of the attention matrix forms a probability distribution over distances (Ben-Kish et al., 2024), as they lie in the $(L - 1)$-simplex (*i.e.* the rows sum to 1). In contrast, expressing a closed-form mapping between input and output tokens for $y = \text{MS-SSM}(x) = f(x)$ is not straightforward. Therefore, we rely on the Jacobian of the output with respect to the input to describe how the sequence is transformed by a MS-SSM layer. We define the *mean mixing distance* for MS-SSM as:

$$d(m, n) = \sum_{n=1}^{m} \frac{|J(x)_{m,n}|}{|\sum_{k=1}^{m} J(x)_{m,k}|} \times (m - n) \tag{5}$$

As the results in Table 6 highlights, MS-SSM achieves a significantly higher mean mixing distance than Mamba, indicating its superior ability to attend to distant contexts, thereby capturing long-range dependencies in the sequence more effectively.

## 4 Conclusion and Discussion

In this paper, we introduced MS-SSM, a multi-resolution state-space model for sequence modeling that integrates multi-scale analysis into state space models (SSMs). By decomposing the system into multiple time scales and incorporating independent SSMs at each resolution, MS-SSM is able to capture dependencies at varying levels of granularity, addressing a key challenge in long-range sequence modeling. The use of specialized convolutions and scale-specific parameter initialization enhances the model's ability to efficiently handle both local and global temporal dynamics.

Our extensive experiments across multiple benchmarks, including image classification, hierarchical reasoning, long-context tasks, and time series tasks, demonstrate the effectiveness of the proposed approach. MS-SSM consistently outperforms state-of-the-art SSM architectures, such as Mamba and Griffin. The results in the Long Range Arena benchmark further validate that MS-SSM can handle effectively long-range dependencies, showing significant improvements over similar data-dependent SSM models. One of the key strengths of MS-SSM lies in its parallelized implementation and minimal computation and model

---

[4]For simplicity, we assume the value projection is $V = x$.

parameters increase, which ensures computational efficiency despite the increased capacity in capturing multi-scale structures.

While MS-SSM is highly effective in capturing multi-resolution and long range dependencies, there remain several avenues for future research. First, extending the MS-SSM framework to other sequence domains, such as natural language processing, where hierarchical structures are prevalent, could further validate its generality. Another potential direction is the exploration of multi-resolution in the most recent form of RNN such as LRU (Orvieto et al., 2023) and xLSTM (Beck et al., 2024) and analyze how it improves the system's memory in such RNN/SSM models.

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

# A Details

## A.1 Notation definition

| Notations | Brief definition and interpretation |
|---|---|
| $x_t, y_t, W$ | the sequence $x \in \mathbb{R}^L$ and $y \in \mathbb{R}^L$ are input and output of a layer, while matrices are denoted by bold uppercase letters, such as layer weight matrix $W$. |
| $\Delta, \bar{A}, \bar{B}$ | discretization step size and parameters of the discrete SSM: $\bar{A} = \exp(\Delta A)$ (state transition matrix), $\bar{B} = (\Delta A)^{-1}(\exp(\Delta A - I)) \cdot \Delta B$. |
| $\hat{x}_t^s, A^s$ | the superscripts denotes the index of a scale: $s \in \{0, \ldots, S+1\}$. $\hat{x}_t^s$ is the $s$-th scale representation of $x_t$, and $A^s$ is the SSM parameter applied to that scale. |
| $[h^0 ; \ldots ; h^{S+1}]$ | Concatenation vectors $\{h^0, \ldots, h^{S+1}\}$ |
| $C_t = \text{Linear}_C(x_t)$ | input-dependent parameter modeled by $\text{Linear}_C(x_t) = W_C x_t$. |
| $\text{Conv1d}(1, 2, L, 2^{s-1})$ | a causal depthwise 1D convolution ($\text{Conv1d}$) with two output channels, a kernel length of $K$, and a dilation factor of $2^{s-1}$ applied to each feature dimension. |
| $h * x$ | linear convolution: $y[t] = (h * x)[t] \triangleq \sum_{\ell=0}^{L-1} h[t-\ell]x[\ell]$ |
| $h \odot x$ | element-wise multiplication (Hadamard product): $y[t] = (h \odot x)[t] \triangleq h[t] \cdot x[t]$ |
| $\text{diag}(A), \text{diag}(a)$ | $\text{diag}(A)$: a vector containing the diagonal elements of square matrix $A$, and $\text{diag}(a)$: a square matrix formed by the entries of $a$ on its diagonal. |
| $\text{Softplus}(.)$ | the nonlinearity defined as: $\log(1 + \exp(.))$ |
| $\text{softmax}(\mathbf{u})$ | Softmax activation function defined as: $\text{softmax}(\mathbf{u})_i := \frac{\exp(u_i)}{\sum_{j=1}^{L} \exp(u_j)}$ |

## A.2 Model Architecture

### A.2.1 Scale Mixing

We explored differnt approaches for scale mixing within the proposed architecture: (i) a data-dependent scale mixing module, as defined in equation 4, (ii) a simple trainable linear layer for scale mixing that is data-independent, and (iii) a data-dependent scale mixing module, similar to the one in equation 4, but uses non-linearity in its gating, expressed as $E_t = s_E(x_t) = \text{SoftMax}(\text{Linear}_E(x_t))$.

The ablation study results, reported in Table 5, indicate that the data-dependent scale mixing with the linear parameterization from equation 4 achieves the best performance among these methods.

## A.3 Effective Receptive Field

We introduce the concept of the *mean mixing distance* as a metric to quantify the effective receptive field (ERF) in our model, drawing inspiration from the receptive field in convolutional networks. This definition is inspired by the average attention distance defined in self-attention models (Dosovitskiy et al., 2020).

For a length-$L$ sequence of tokens $\mathbf{x} = (x_1, x_2, \ldots, x_L)$, the self-attention layer transforms the sequence by computing a weighted sum of token embeddings, as follows:

$$\mathbf{y} = \text{SA}(\mathbf{x}) = \text{SoftMax}\left(\frac{\mathbf{Q}\,\mathbf{K}^T}{\sqrt{d_k}}\right)\mathbf{x} = \mathbf{A}(x)\,\mathbf{x},$$
$$\text{where} \quad \mathbf{Q} = \mathbf{x}\,\mathbf{W}_Q, \quad \mathbf{K} = \mathbf{x}\,\mathbf{W}_K,$$

In this equation, the matrix $\mathbf{A}(x)$ contains the normalized attention scores between each pair of tokens. Which defines the mapping between each output token and all tokens in

Table 6: Comparison of Mean Mixing Distance between Mamba and MS-SSM on the ListOps dataset. The metric $d(m, L)$, as defined in (6), is averaged across all channels and layers in the model.

| Method | Mean Mixing Distance |
|---|---|
| Mamba | $38.84_{\pm 21.97}$ |
| MS-SSM (S6) | $94.90_{\pm 64.62}$ |

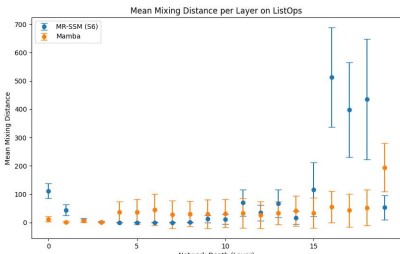

the input sequence.[5] Using this, the average attention distance (Dosovitskiy et al., 2020) is defined as:

$$d(m, n) = \sum_{n=1}^{m} \mathbf{A}(x)_{m,n} \times (m - n)$$

where each row of the attention matrix forms a probability distribution over distances (Ben-Kish et al., 2024), as they lie in the $(L - 1)$-simplex (*i.e.* the rows sum to 1).

In contrast, expressing a closed-form mapping between input and output tokens for $\mathbf{y} = \texttt{MS-SSM}(\mathbf{x}) = f(\mathbf{x})$ is not straightforward. Therefore, we rely on the Jacobian of the output with respect to the input to describe how the sequence is transformed by a MS-SSM layer. The Jacobian matrix defined as the collection of the gradient of each output token with respect to the input sequence: $\mathbf{J}_f = \begin{bmatrix} \nabla^{\mathrm{T}} f_1 \\ \vdots \\ \nabla^{\mathrm{T}} f_L \end{bmatrix}$. We define the *mean mixing distance* for MS-SSM as:

$$d(m, n) = \sum_{n=1}^{m} \frac{|\mathbf{J}(x)_{m,n}|}{|\sum_{k=1}^{m} \mathbf{J}(x)_{m,k}|} \times (m - n) \tag{6}$$

where the Jacobian is normalized row-wise to form a probability distribution over the distance analogous to attention-based models. In classification tasks, we compute $d(m, L)$, the mean mixing distance for the last token, as a measure of the ERF, capturing how far dependencies extend across the sequence in MS-SSM.

As the results in Table 6 highlights, MS-SSM achieves a significantly higher mean mixing distance than Mamba, indicating its superior ability to attend to distant contexts, thereby capturing long-range dependencies in the sequence more effectively.

## A.4 Efficient Implementation of Multi-scale Decomposition Layer.

While computation of multi-scale decomposition (2) requires sequential application of a convolution layer, this filtering scheme is actually linear time-invariant (LTI) and can be implemented using linear convolution layers. Composing two linear convolution layers $\varphi_1$ and $\varphi_2$ with kernel sizes $K_1$ and $K_2$, respectively, yields a single linear convolution layer $\varphi_{1:2} = \varphi_1 * \varphi_2$ with an effective kernel size of $K_1 + K_2 - 1$. This property enables us to transform this sequential linear convolutions into a parallel application of array of filter banks during inference. When the filter length and number of levels are limited, this approach can potentially accelerate multi-resolution decomposition by leveraging specialized implementations of convolution units available on modern hardware accelerators, resulting in a more hardware-efficient solution.

---

[5]For simplicity, we assume the value projection is $V = \mathbf{x}$.

# B Experimental Details

For all the experiments, we use the same experimental setup as Smith et al. (2023) and Shi et al. (2023). The results of baselines are either from the original papers, or are reported by Shi et al. (2023) and/or Qin et al. (2024).

## B.1 Image Classification

We employ the Vision Transformer (ViT) architecture (Dosovitskiy et al., 2020), integrating MS-SSM as the core block. The models are evaluated on two image classification tasks: sCIFAR (Shi et al., 2023) and ImageNet-1K (Krizhevsky et al., 2012).

**sCIFAR-10:** For the sCIFAR-10 dataset, each image is transformed into a sequence of pixels with size 1024 and 3 channels, and the model is built using a ViT architecture (Dosovitskiy et al., 2020) consisting of 10 layers with a hidden size of 256 and filter size of 2. The Adam optimizer with standard settings ($\beta_1 = 0.9, \beta_2 = 0.999$) and a learning rate of 0.0045 was used, along with a linear warmup over the first 1 epoch. A weight decay of 0.01 was applied as regularization. We use $S = 3$ and $N = 128$. The model was trained on A6000 GPUs for 250 epochs with a batch size of 50.

**ImageNet-1K:** In the case of ImageNet-1K, images were divided into patches of $16 \times 16$ pixels, and we trained a ViT-base architecture (Dosovitskiy et al., 2020) with 24 layers and a hidden size of 256. Training was conducted using the Adam optimizer with a base learning rate of 1e-3 and its standard settings ($\beta_1 = 0.9, \beta_2 = 0.999$). The learning rate scheduler included a linear warmup for the first 10 epochs, followed by a cosine decay. MS-SSM was trained for 300 epochs using 4xA6000 GPUs with a batch size of 1024. Each MS-SSM layer consists of a multi-scale convolution with $S = 3$ scales, each convolution having a length of $K = 4$, and SSMs with a latent state size of $N = 128$.

**ListOps:** We use the setting of Long-range Arena (Tay et al., 2020b) benchmark and pad all sequences to the length of 2048 and then use an embedding layer to encode them into 128 channels. We use 20 layers of MS-SSM to mach the number of parameters of other models in the benchmark study. In MS-SSM we choose filter size as 4 and dimension of 128. The model is trained for 100 epochs with batch size of 50. Following Shi et al. (2023), we use AdamW optimizer with a weight decay rate 0.03, learning rate of 0.003 after 1 epoch of linear warmup, and a dropout rate 0.1. The batch normalization is used instead of layer normalization. We use $S = 3$ and $N = 128$.

**Long Range Arena:** We use the settings from Long Range Arena benchmark (Tay et al., 2020b) but to match the number of parameters, we use $\times 2$ of the number of layers for Transformers.

**PTB-XL** In this dataset, we have 12 channels, each of which has 1000 timestamps. All the architectural setting for this experiment is the same as the CIFAR10 but instead of batch normalization, we use layer normalization. We use dropout rate of 0.2 and the AdamW optimizer with weight decay rate 0.06. The network is train for 5 warmup epochs and then 95 epochs of cosine learning rates.

# C Additional Experiments and Ablations

## C.1 Ablations

In this section, we compare our initialization with the Mamba's initialization. The results are reported in Table 7. As expected, the scale-dependent initialization scheme proposed in this work is more effective and MS-SSM achieve better performance when using such initialization.

Table 7: Ablation studies on the initialization of MS-SSM.

| | Method | PTB-XL | ListOps |
|---|---|---|---|
| | Base | | |
| 1 | MS-SSM | 0.939 | 63.04 |
| 2 | Mamba's Initialization | 0.928 | 57.49 |

