# OpenReview forum: "MS-SSM: A Multi-Scale State Space Model for Efficient Sequence Modeling"
_colmweb.org/COLM/2025/Conference — COLM 2025_

### Official Review · Reviewer_Rtcz · 2025-05-05

**Rating:** 7
**Confidence:** 3
**Ethics Flag:** 1

**Summary:**

This paper introduces MS-SSM, a multi-scale state space model for efficient sequence modeling. The method decomposes the input into multiple temporal resolutions using a wavelet-inspired convolutional layer, applies a separate SSM at each scale, and combines their outputs through an input-dependent mixing mechanism. This design aims to address the limitations of standard SSMs in modeling long-range and hierarchical dependencies. The model is evaluated on a diverse set of tasks including image classification, time series classification, and hierarchical reasoning, achieving consistent gains over prior SSM-based models such as S4 and S6, and outperforming Mamba in several benchmarks. Ablation studies highlight the importance of both the multi-scale decomposition and the scale mixer.

**Questions To Authors:**

- Extension to selective scan models like Mamba
- Evaluation on long-context NLP benchmarks

**Reasons To Accept:**

1. The multi-scale decomposition addresses known limitations of standard SSMs in modeling long-range and hierarchical structure, and the overall architecture is clean, modular, and intuitive.

2. The model consistently outperforms prior SSM-based baselines on a diverse set of tasks including vision, time series, and structured reasoning.

3. The paper carefully isolates the contributions of the multi-scale convolution and the scale mixer, providing insight into which components are most important for performance.

4. The paper is well-organized and easy to follow, with sensible baselines and reproducible experimental details.

**Reasons To Reject:**

**1. Limited applicability to selective scan models**

The proposed method is evaluated only with S4 and S6, which use standard recurrence or input gating. However, recent state-space models such as Mamba rely on selective scan, where recurrence parameters are computed from the input at each position. Applying the multi-scale framework to such models may require running a separate selective scan per scale, which increases memory usage, compute cost, and may lead to training instability. The paper does not address how MS-SSM could be extended to this class of models, leaving open questions about its compatibility with the latest SSM architectures.

---
**2. No evaluation on long-context NLP tasks**

The paper focuses on improving long-range and hierarchical sequence modeling, but it does not include any experiments on natural language (NLP) tasks where these challenges are most critical. Long-context benchmarks such as lost-in-the-middle are commonly used to evaluate this ability. Without such evaluations, it is difficult to assess the model’s effectiveness in NLP settings where long-context reasoning plays a central role.

---

> ### Author Response · Authors · 2025-06-03
>
> We sincerely thank the reviewer for their thoughtful evaluation and constructive feedback. We address the reviewer’s concerns below.
>
> > 1. Limited applicability to selective scan models
>
> We would like to clarify that the MS-SSM (S6) implementation used in our experiments does, in fact, *use Mamba-style input-dependent parameterization as the underlying SSM*. Specifically, in our architecture, the parameters of each SSM at each scale are computed based on the original input $x_t$ (detailed in the subsection “Input-Dependent Parameterization”)​, mirroring the selective scan mechanism introduced in Mamba. We did not observe any stability issues during training. As stated in Section 2.2, we maintain a consistent overall state size by setting the effective state dimension of our multi-scale array—$(S + 2) \times N$—equal to the recurrent state size of the baseline Mamba model.
>
> >2. No evaluation on long-context NLP tasks
>
> We agree that extending our evaluations to NLP tasks is an interesting next step. For this initial paper, our goal was to validate the core hypothesis of the MS-SSM framework across a diverse set of foundational domains where long-range and hierarchical dependencies are critical such as Long Range Arena (LRA) and hierarchical reasoning tasks like ListOps. As noted in our conclusion, we view NLP benchmarks, such as language modeling, as important future applications to further validate the generality of MR-SSM.

---

### Official Review · Reviewer_x51h · 2025-05-09

**Rating:** 6
**Confidence:** 3
**Ethics Flag:** 1

**Summary:**

This paper proposes MS-SSM, a multi-scale state-space model (SSM) for efficient sequence modeling. It enhances traditional SSMs by processing sequences at multiple resolutions, combining specialized dynamics with a learnable scale mixer to capture both fine-grained and coarse patterns. The authors claim improved memory efficiency and long-range modeling, supported by experiments on Long Range Arena, time series, and image tasks.

**Reasons To Accept:**

The multi-scale SSM framework is a novel extension, with innovations like the scale mixer and specialized dynamics. However, the significance is limited by the lack of consistent performance gains across all benchmarks. More rigorous comparisons with non-SSM baselines (e.g., attention, RNNs) would better demonstrate its advantages. The concept is promising, but the empirical validation needs strengthening.

**Reasons To Reject:**

1. The paper focuses primarily on accuracy, but efficiency in sequence modeling involves more than just accuracy-speed, memory usage, and scalability are equally important. The experiments lack comparisons on these aspects, weakening the claim of "efficient" modeling. Additionally, the performance gains are inconsistent: Table 1 shows marginal improvements, while Table 2 reports large gains. Without robustness and generalization analysis, the overall impact remains unclear.
2. The motivation is too broad—while multi-scale modeling is important, the specific gaps in SSMs are not sharply defined. A clearer justification for why existing SSMs fail at multi-scale tasks would strengthen the contribution. The writing is clear, but the experimental results need deeper discussion to explain the inconsistent improvements across tasks.

---

> ### Author Response · Authors · 2025-06-02
>
> We sincerely thank the reviewer for their thoughtful and constructive feedback. We address your concerns below and will revise the final version of the paper accordingly.
>
> > More rigorous comparisons with non-SSM baselines (e.g., attention, RNNs) would better demonstrate its advantages.
>
> We would like to gently clarify that our comparison tables also include a range of non-SSM baselines (e.g., attention, RNNs). All experiments report results for Transformer-based models, and Table 1 includes various RNNs (LSTM, HIPPO-RNN, GRU etc.) and Convolution baselines, Table 2 includes an extensive list of Transformer variants (BigBird, Longformer, etc.), Table 3, 4 also include RNN baselines (e.g. LSTM and LRU). While these results provide broad baselines, we chose to focus our analysis on the comparison with state-of-the-art SSMs like Mamba, as our contribution focuses on architectural improvements to these SSM (RNN) architectures.
>
> > The paper focuses primarily on accuracy, but efficiency in sequence modeling involves more than just accuracy…
>
> We appreciate the reviewer's point on efficiency. In our paper, we refer to “efficiency” in terms of asymptotic complexity. MS-SSM maintains the same favorable complexity and scalability as the underlying SSMs (e.g., Mamba), with linear-time inference and parallelizable training. As discussed in the "Complexity" subsection, our multi-scale convolution layer adds only a minimal overhead of $\mathcal{O}(LKS)$ in computation and $\mathcal{O}(KS)$ in parameters per layer, where $L$ is the sequence length, $K$ is the kernel size, and $S$ is the number of scales. In practice, we  typically use a small kernel size $K \in [3, 5]$ and set $S=3$, keeping this overhead negligible.
>
> To further support our claim of *memory efficiency*, we introduce the *mean mixing distance* (Section 3.1) as a measure of the *effective receptive field (ERF)*. As reported in Table 6, MR-SSM achieves a significantly higher ERF than Mamba, indicating its stronger ability to attend to long-range dependencies.
>
>
>
> > Additionally, the performance gains are inconsistent: Table 1 shows marginal improvements, while Table 2 reports large gains.
>
> We thank the reviewer for highlighting this observation across different tasks. We see this not as a weakness, but as an important finding that validates our core hypothesis: *multi-scale modeling is most impactful in tasks with long-range or hierarchical dependencies*.
> *Tasks with Large Gains:* The substantial improvements are seen on tasks that are inherently hierarchical (ListOps in Table 2, where MS-SSM achieves a significant gain over Mamba ($63.04$% vs. $38.02$%) or require modeling very long-range dependencies (LRA in Table 4. MS-SSM achieves $14.42$% absolute improvement over Mamba).
> *Tasks with Smaller Gains:*  Image classification tasks such as sCIFAR, ImageNet-1K in Table 1, are included to demonstrate the generalizability of MR-SSM beyond standard sequence benchmarks. Given the already high baseline accuracy in these tasks, achieving large improvements is inherently more difficult. Nevertheless,  MR-SSM still achieves consistent improvement over Mamba.
>
> > A clearer justification for why existing SSMs fail at multi-scale tasks would strengthen the contribution. The writing is clear, but the experimental results need deeper discussion
>
> Thank you for this suggestion. As discussed in the motivation of the paper, SSMs have limited, effective memory requiring larger state sizes for improved recall. We support this by introducing the mean mixing distance metric, which quantifies the effective receptive field in Experiments section and we empirically show that MS-SSM achieves a significantly higher ERF than Mamba  in Table 6.
>
> We hope the clarification above addresses your concerns.

---

### Official Review · Reviewer_FfTf · 2025-05-10

**Rating:** 7
**Confidence:** 3
**Ethics Flag:** 1

**Summary:**

This paper discuss the multi-scale state-space models in the sequence modeling task.

**Questions To Authors:**

NA.

**Reasons To Accept:**

1. Extensive numerical experiments are conducted, including classical sequential CIFAR, LRA, and ECG benchmarks. An ablation study is also provided to illustrate the contribution of each component.
2. The metric proposed in Section 3.1 is an interesting concept. Although its relationship with attention mechanisms and practical optimization applicability is underexplored, introducing new metrics is valuable for enhancing the understanding of sequence models.

**Reasons To Reject:**

1. The theoretical connection between the multi-scale argument and the model's final performance is limited, as the discussion mainly focuses on input-dependent parameterization for improved multi-scale approximation. It is recommended to extend the theoretical analysis to parameterization theorems, similar to the theorem presented in StableSSM (https://arxiv.org/abs/2311.14495). If it's not achievable, maybe it's better to demonstrate a theorem to show that the improvement from multi-scale resolution is significant even when the models get larger.
2. A minor issue is the absence of substantial NLP tasks for language modeling and downstream evaluation. Although these tasks are computationally demanding, it is still advisable to train a reasonably scaled model (e.g., 120M parameters) for comparison with GPT-2, Mamba, and GLA.

---

> ### Author Response · Authors · 2025-06-03
>
> We sincerely thank the reviewer for the thoughtful feedback. Below, we address the reviewer's suggestions and concerns in detail.
>
> >It is recommended to extend the theoretical analysis to parameterization theorems, similar to the theorem presented in StableSSM
>
>  We appreciate the reviewer’s interest in a stronger theoretical justification. Our approach to architectural design is motivated by well-established signal processing theory (e.g., Stationary Wavelet Transform), which supports the use of multi-resolution representations for capturing long-range structure.
> We agree that a theoretical analysis of the parameterization power of multi-scale SSMs, similar to what StableSSM explores for continuous-time linear time-invariant (LTI) SSMs, would be valuable. However, extending such theoretical tools to multi-resolution settings—especially when combined with data-dependent parameterizations—poses substantial technical challenges and is an exciting direction for future work.
>
> That said, we do attempt to bridge the theoretical-experimental gap through:
> - the *mean mixing distance* introduced in Section 3.1, which quantifies *the effective receptive field* and supports our hypothesis about multi-scale modeling,
> - and comparisons with larger models (e.g., Mamba with 2× state size and parameter count), where MR-SSM still outperforms, suggesting that its gains are not simply due to scaling capacity but to improved inductive bias in capturing long hierarchical structures.
>
> Moreover, our *scale-dependent initialization scheme* is a practical application of the theory connecting an SSM's eigenvalues to its effective memory (Agarwal et al., 2023). That is, SSMs for coarse-grained scales are initialized with eigenvalues closer to 1, prioritizing long-range memory needed to capture global trends, while fine-grained scales are initialized with smaller eigenvalues to prioritize shorter effective memory.
>
> > A minor issue is the absence of substantial NLP tasks for language modeling
>
> We agree that extending our evaluations to NLP tasks is an interesting direction. In this work, our main focus was to assess the core capability of multi-resolution SSMs on long-context modeling and hierarchical structure learning, which is directly measured via Long Range Arena (LRA) and hierarchical reasoning tasks such as ListOps. As noted in our conclusion, we view NLP benchmarks, such as language modeling, as important future applications to further validate the generality of MR-SSM as it is fully compatible with causal language modeling architectures.

---

### Official Review · Reviewer_xBk6 · 2025-05-13

**Rating:** 7
**Confidence:** 3
**Ethics Flag:** 1

**Summary:**

This work introduces an extension to the State Space Model (SSM) by adding multi-scale convolution layer. The motivation for this additional layer is to capture multi-scale dependencies in order to better model complex structures.
First, the input signal is decomposed into multiple scales and later combined with a scaler-mixer. The additional components introduce a small overhead in terms of parameters and computation.
Experiments were conducted on well-known, SSM-related tasks such as sCIFAR, ImageNet-1K and Long Arena benchmark. The introduced model yields superior results over Mamba and other models.

This work is well motivated and easy to follow. The experimental setup is quite vanilla and this paper could be even more convincing by providing results on a range of popular (NLP-related) downstream tasks (similar to (Gu & Dao, 2023)).
Furthermore, it would be great to quantity “minimal computation and model parameters increase”.

**Questions To Authors:**

- Can you quantity “minimal computation and model parameters increase”?
- Please highlight best (and second best) results in Table 4

**Reasons To Accept:**

- Well written and motivated
- Simple and effective method to extend SSMs

**Reasons To Reject:**

- Vanilla SSM-related experimental setup only

---

> ### Author Response · Authors · 2025-06-02
>
> We appreciate the reviewer for the positive feedback and for recognizing the simplicity and effectiveness of our proposed approach. We address the questions and suggestions below:
>
> > Can you quantity “minimal computation and model parameters increase”?
>
> The additional computation primarily stems from the 1D dilated convolutions and scale mixing MLPs, which are lightweight and parallelizable. As stated in the "Complexity" paragraph at the end of Section 2, the multi-scale convolution operation adds a computational overhead of $\mathcal{O}(LKS)$ and $\mathcal{O}(KS)$ parameters per layer, where $L$ is the sequence length, $K$ is the kernel size, and $S$ is the number of scales. To make this concrete, for most of our experiments (e.g., on ImageNet-1K, sCIFAR, ListOps), we typically used a small kernel size of $K \in [3, 5]$ and $S=3$ scales, which introduces only a marginal increase in parameter count.  We will highlight these quantitative parameters and computational overhead in the appendix.
>
> > Please highlight best (and second best) results in Table 4
>
> Thank you for the suggestion. We will update Table 4 to visually highlight both the best and second-best results for each task in bold and underline, respectively, for improved readability. We also wish to clarify the main takeaway from this table. As noted in the footnote, our primary goal was not necessarily to achieve state-of-the-art results across the entire LRA benchmark, but to conduct a fair and direct comparison against a similar SSM-based architecture, Mamba. On the LRA benchmark average, MS-SSM achieves a $14.42$% absolute improvement over Mamba ($86.73$% vs. $72.30$%).
>
> **Additional Note:**
> We agree with the reviewer that extending our evaluations to NLP-related downstream tasks is an interesting future direction. In this work, our primary focus was to assess the core capability of multi-resolution SSMs on long-context modeling and hierarchical structure learning, which is directly measured via Long Range Arena (LRA) and hierarchical reasoning tasks such as ListOps. As noted in our conclusion, we view NLP benchmarks, such as language modeling, as important future applications to further validate the generality of our method.

---

> > ### Comment · Reviewer_xBk6 · 2025-06-04
> >
> > Regarding
> > > Can you quantity “minimal computation and model parameters increase”?
> >
> > I was actually hoping for concrete numbers based on the models employed in your experimental setup.

---

### Decision · Program_Chairs · 2025-07-08

**Decision:**

Accept

**Comment:**

This paper introduces MS-SSM, a multi-scale state-space model framework that addresses limitations in traditional SSMs by processing sequences at multiple resolutions with specialized dynamics and an input-dependent scale mixer. The work presents a well-motivated and technically sound approach to enhance SSMs' ability to capture both fine-grained patterns and coarse global trends, demonstrating consistent improvements across diverse benchmarks including Long Range Arena, time series classification, and image recognition tasks. The multi-scale decomposition is intuitive and addresses known limitations of standard SSMs in modeling hierarchical dependencies. The experimental evaluation is comprehensive with proper ablation studies that isolate the contributions of key components, and the writing is clear and well-organized throughout.

Pros: Well-motivated multi-scale framework addressing clear limitations in existing SSMs; consistent performance improvements across diverse tasks (vision, time series, structured reasoning); comprehensive ablation studies demonstrating the value of each component; clean, modular architecture that integrates naturally with existing SSM designs.

Cons: Limited evaluation on natural language processing tasks, particularly long-context NLP benchmarks where multi-scale modeling would be most critical. The work could be improved with experiments in standard language-model tasks.